# Comparison of Intestinal Permeability Methods in Broilers over a 6-Week Growth Period

Maddison L. Wiersema [1], Brian J. Kerr [2] and Dawn A. Koltes [1],*

1 Department of Animal Science, Iowa State University, Ames, IA 50011, USA; wiersema@iastate.edu
2 USDA-ARS National Laboratory for Agriculture and the Environment, Ames, IA 50011, USA; brian.kerr@usda.gov
* Correspondence: delkins@iastate.edu

**Abstract:** The adoption of methods detecting intestinal permeability in poultry has been slow due to the lack of urine availability in avian species. The objective of this study was to examine intestinal permeability assays in broilers using serum. Fluorescein isothiocyanate-dextran (FITC-D) and lactulose/mannitol/sucralose (LMS), indigestible sugars, were used to detect intestinal permeability across two fed states (fed or fasted) and four sugar treatments (Control, FITC-D, LMS, or FITC-D+LMS). Broilers housed in pens were assigned one of eight treatments and sampled on 14, 28, and 42 days of age. Data were analyzed using PROC Glimmix for fed state, sugar treatment, age, and all interactions. Serum lactulose and FITC-D increased in fasted compared to fed birds ($p < 0.006$), whereas mannitol increased in fed compared to fasted birds ($p < 0.001$). Serum lactulose and FITC-D decreased on day 28 compared to other timepoints ($p < 0.003$). Serum FITC-D only had a significant sugar by fed state interaction ($p < 0.05$) with elevated concentrations in fasted and fed birds that received FITC-D. Serum lactulose was significant for all interactions with elevated concentrations in broilers provided lactulose and fasted ($p < 0.001$). The ability to detect a three-way interaction with serum lactulose suggests an increased sensitivity; however, additional studies are needed.

**Keywords:** fasted; fluorescein isothiocyanate-dextran; lactulose; mannitol; serum





## 1. Introduction

The need to better understand intestinal permeability is growing in the livestock industry as producers seek to prevent or reduce the magnitude of enteric disease. Normally, the intestinal barrier is selectively permeable and controls the passage of water, nutrients, electrolytes, and other essential ions [1]. Many factors such as age, diet, shifts in intestinal microbiota, infection, disease, environment, and stress can impact the intestinal barrier, leading to increased permeability [2]. Increasing the rate of non-mediated passive diffusion or intestinal permeability, through or between intestinal epithelial cells, can lead to decreased performance, enteric disease, and systemic disease [2,3]. Increased intestinal permeability through different stressors in poultry may lead to decreased production and performance, compromised health, and lameness [4]. It is known that stressors, such as toxins, parasites, dysbiosis, and environmental stressors (heat stress and feed restriction) can also be causes of intestinal stress leading to disruption of the intestinal tract [1,4–9]. In broilers, it has been observed that increased intestinal permeability can lead to poor feed conversion, lower body weights (BW), and increased incidences of lameness and other skeletal diseases [2,4]. Therefore, identifying a consistent method to determine intestinal permeability in non-disease states that can be used on commercial farms is critical for understanding and improving intestinal health and overall performance of poultry.

A variety of in vivo methods for measuring intestinal permeability are available for mammals. In human medicine and a variety of animal species, such as dogs, rats, and pigs, sugars such as lactulose, mannitol, rhamnose, and sucralose are used in dual or multi-sugar tests to measure intestinal permeability in urine [2,10–12]. The variation in size

and molecular weight of these sugar molecules allows researchers to better estimate the degree of permeability by measuring the relative amounts of each sugar that has passed through the intestinal barrier into the bloodstream then into the urine. It is thought that larger molecules such as lactulose and sucralose are more likely to move paracellularly through the tight junctions, while monosaccharides such as rhamnose and mannitol move transcellularly through the enterocytes at a more uniform rate [13]. Because avian species lack the excretion of urine, this method has only been sparsely explored in poultry [14]. Another in vivo method of measuring intestinal permeability is the use of fluorescein isothiocyanate-dextran (FITC-D), which is a non-digestible, fluorescently labeled dextran molecule that has been validated for measuring intestinal permeability in poultry and is widely used in other species [15–18]. The size of FITC-D is too large to pass through the intestinal barrier under normal conditions and moves from the lumen into the bloodstream via passive paracellular transport during times of stress, inflammation or infection [17]. While both methods have been used as biomarkers to assess intestinal permeability in poultry, it is crucial to directly compare these methods and determine a method that can be used to measure intestinal permeability in a commercial setting. Therefore, the primary objective of this study was to compare these two different methods, serum FITC-D and serum lactulose, mannitol, and sucralose (LMS) in measuring intestinal permeability in poultry in the presence or absence of a known stressor over the 6-week production cycle in broilers.

## 2. Materials and Methods

All procedures involving live animals were approved by the Iowa State University Institutional Animal Care and Use Committee (IACUC number 18–331). Two independent but consecutive trials were conducted from March to June 2019 at the Iowa State University Poultry Teaching and Research Farm. The experimental design was the same for both trials with the only difference being the genetic strain of bird used due to availability of lines at a local commercial hatchery. Broiler chicks ($n = 500$ per trial) were obtained from a local commercial hatchery (Whelp Hatchery, Bancroft, IA, USA) and transported to the Iowa State University Poultry Teaching and Research Farm. Chicks were commonly raised for the first 7 days. At 7 days of age, 480 chicks were weighed and randomly allocated to 40 ($1.2 \times 1.2$ m) floor pens with 12 birds per pen. Pens were the experimental unit and designated to 1 of 2 fed states (fed or fasted) and 1 of 4 sugar treatments (FITC-D only, LMS only, FITC-D+LMS (FITC-LMS), or water). All pens were sampled at 3 time points (14, 28, and 42 days of age (doa)) which resulted in 5 pens per fed state and sugar treatment at each time point. See Figure 1 for a detailed pen layout of the experimental design. Typical management strategies were followed regarding temperature and lighting schedules. Birds were allowed ad libitum access to feed via hanging feeders, except for pens designated as fasted, which had feeders removed 12 h before the administration of sugars. Diets can be found in Supplemental Table S1. Birds were allowed ad libitum access to water via a nipple drinker system at all times. The same starter (day 0–14), grower (day 15–28), and finisher (day 29–42) diets were provided to all birds with sampling of the birds occurring on the last day of each diet phase.

### 2.1. Intestinal Permeability

During each 6-week trial, three sample collection days occurred. Twelve hours before sampling, feed was removed from pens designated as fasted ($n = 20$). On days of sample collection, pen weights were collected and used to determine gavage-treatment dosage. On average, 4 birds were randomly chosen from each pen and orally gavaged with 1 mL of the designated treatment for the pen: FITC-D only, lactulose, mannitol, and sucralose (LMS) only, FITC-D+LMS (FITC-LMS), or control/no sugar treatment (water). Due to high mortality during the first trial, the number of biological trials was reduced in 11 of the 40 pens, which had less than 4 birds at various sampling days, whereas only 5 of the

40 pens had less than 4 birds in the second trial. These mortalities during the first 7 days were primarily a result of poor starting chicks or yolk sac infections.

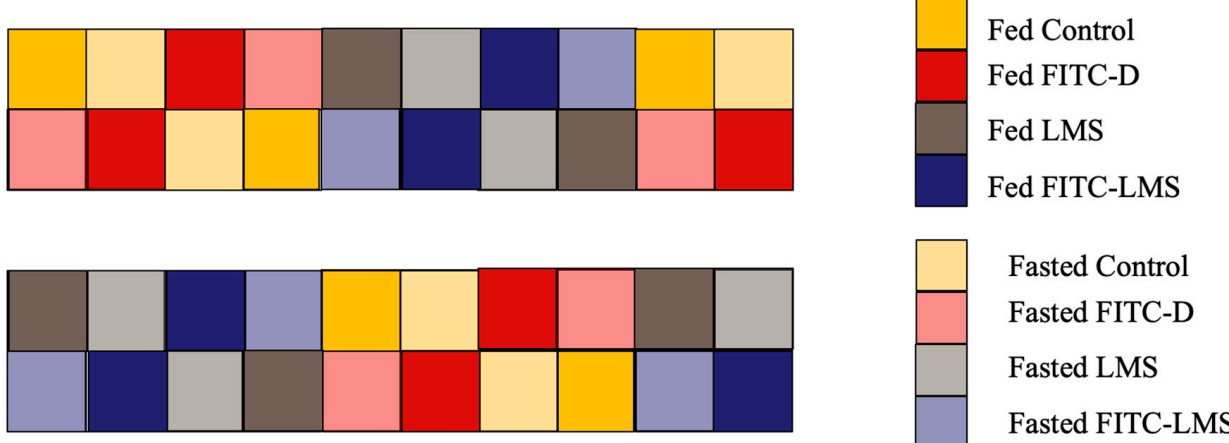

**Figure 1.** Representative experimental layout of the broiler floor pens. Each box represents a single pen with the color of the pen corresponding to the treatment (sugar treatment and fed status). The experimental layout was replicated with fed and fasted pens altering at each time point. Sugar treatments include the control group, which was provided water; FITC-D group, which was provided fluorescein isothiocyanate-dextran at a rate of 8.32 mg FITC-D/kg BW; LMS group, which was provided lactulose, mannitol, and sucralose at a rate of 0.25 g lactulose/kg BW, 0.05 g mannitol/kg BW, and 0.05 g sucralose/kg BW; and the FITC-LMS group was provided with a combination of FITC-D and LMS at rates described above.

### 2.2. Sugar Treatments

FITC-D with an average molecular weight of 3000 to 5000 (FD4, Sigma Aldrich, St. Louis, MO, USA) was dissolved in distilled water and administered at a rate of 8.32 mg FITC-D/kg BW [17]. A mixture of lactulose, mannitol, and sucralose (Sigma Aldrich, St. Louis, MO, USA) was dissolved in distilled water and given at rates of 0.25 g lactulose/kg BW, 0.05 g mannitol/kg BW, and 0.05 g sucralose/kg BW [14]. The combination of FITC-D and LMS was administered at the same rates as stated above, but as a single solution. Water was administered as the control solution. Once orally gavaged, birds were marked with animal-safe paint to denote treatment and returned to their home pen. One hour after oral gavage, birds were euthanized via $CO_2$ asphyxiation and death was confirmed by checking the nictitating eyelid response. Blood samples were collected from all birds from the femoral artery into collection tubes (BD Medical, Franklin Lakes, NJ, USA) and transported back to the laboratory. Blood was allowed to clot at room temperature followed by centrifugation at $1000 \times g$ for 15 min. Following separation, serum was portioned into 0.5 mL aliquots and stored in amber tubes at $-80\ °C$ until analysis.

### 2.3. FITC-D Analysis

Methods for analysis of FITC-D in blood serum were modeled from a previously described and optimized protocol from Baxter et al. [18]. Briefly, serum was analyzed at the conclusion of both trials and serum from control birds was combined and diluted at a 1:5 ratio with phosphate-buffered saline (PBS) and used to create a standard curve. Serum samples from birds that received treatments were also diluted at a 1:5 ratio using PBS. An amount of 100 µL of diluted sample from each bird or standard curve was plated in triplicate on a black 96-well plate. Fluorescence was measured using a BioTek Cytation spectrophotometer (BioTek US, Winooski, VT, USA) with excitation and emission wavelengths of 485 and 528 nm, respectively. Average concentration of FITC-D in serum was calculated as ng/mL per bird. Results were averaged per bird for data analysis.

*2.4. Lactulose, Mannitol, and Sucralose Analysis*

All individual serum samples were analyzed for lactulose and mannitol using a modified method described by Fleming et al. [19] at the conclusion of both trials. In brief, serum samples were diluted at a rate of 1:4 dilution with ultra-pure water. The solution underwent centrifugal ultrafiltration, using a Nanosep filtration device with a 30 K membrane, removing insoluble particles. Samples were then analyzed for blood sugars on an ion chromatograph (Dionex ICX-6000, ThermoFisher Scientific, Waltham, MA, USA) using carbohydrate disposable working electrodes. The average run time for each sample was approximately 60 min. Several samples of lactulose lacked detectable limits in the serum. For analysis, these samples were assigned the arbitrary value of 1. Additionally, sucralose was not detected in any samples and, therefore, sucralose was not included in any additional analysis.

*2.5. Statistical Analysis*

Serum data for each sugar were analyzed using the PROC GLIMMIX procedure in SAS Cary, NC, USA; [20] and analyzed for the fixed effects of day (14, 28, or 42), fed state (fed or fasted), and sugar treatment (FITC-D, LMS, FITC+LMS, or control). All 2- and 3-way interactions between day, fed state, and sugar treatment were analyzed. The replicate (trial 1 and trial 2) was fit as a random effect and the pen (1 through 40) was fit as a repeated measure. The model distribution was set as a lognormal distribution. Residuals were tested for normality and homoscedasticity using Proc Univariate when following a lognormal distribution [20]. Least-squares (LS) means were calculated for all effects using the LSmeans statement and all LSmeans presented were back-transformed for publication. *p*-values for differences of the LSmeans were generated using the pdiff option and the Tukey procedure was performed to account for multiple comparisons. Statistical significance was set at $p < 0.05$.

## 3. Results

*3.1. Animal Parameters*

All broilers used in this study were seemingly healthy at the time of sampling. In the first trial, the average BW at the time of sampling was $0.48 \pm 0.02$ kg, $1.60 \pm 0.02$ kg, and $3.13 \pm 0.02$ kg, at 14, 28, and 42 days of age (doa), respectively. In the second trial, the average BW at the time of sampling was $0.38 \pm 0.01$ kg, $1.40 \pm 0.01$ kg, and $2.84 \pm 0.01$ kg at 14, 28, and 42 doa, respectively. Overall mortality for the first trial was 3.75% (18/480), where most of the mortality occurred within the first 7 days (2.5%) prior to pen assignment. Total mortality for the second trial was 1.87% (9/480). No mortality occurred during the first 7 days of the second trial.

*3.2. Serum Fluorescein Isothiocyanate-Dextran Concentrations*

The effect of bird age, fed state, sugar treatment, and the two-way interaction of fed state and sugar treatment were significantly different for serum FITC-D concentrations ($p < 0.05$). Serum FITC-D concentrations were greater at 14 doa compared to 28 doa ($p < 0.001$; Table 1) with serum FITC-D on 42 doa being similar to 14 and 28 doa ($p > 0.072$). Serum concentrations of FITC-D were greater in fasted broilers compared to fed broilers ($p = 0.004$; Table 2). Serum samples from birds that did not receive FITC-D (control and LMS) were similar to each other ($p = 0.202$) and were decreased compared to the serum from broilers given FITC-D ($p < 0.001$; Table 3). Serum from birds given only FITC-D had greater serum FITC-D concentrations compared to birds given FITC-D+LMS ($p = 0.001$). Additionally, serum from birds given FITC-D+LMS had similar serum FITC-D concentrations as serum from birds that did not receive FITC-D ($p > 0.067$).

**Table 1.** Effect of day on average serum sugar concentrations in broilers during a 42-day production cycle across 2 replicates.

| | FITC-D * | SEM | Mannitol | SEM | Lactulose ** | SEM | L:M | SEM |
|---|---|---|---|---|---|---|---|---|
| **Day** | **ng/mL** | | **µg/mL** | | **µg/mL** | | | |
| 14 | 122.86 [a] | 8.92 | 7.51 [a] | 1.82 | 2.75 [a] | 0.11 | 0.83 [b] | 0.06 |
| 28 | 48.39 [c] | 8.89 | 6.24 [b] | 1.51 | 2.76 [a] | 0.11 | 0.91 [a] | 0.07 |
| 42 | 99.22 [b] | 9.03 | 5.71 [b] | 1.39 | 2.32 [b] | 0.10 | 1.09 [a] | 0.08 |
| *p*-value | 0.003 | | 0.002 | | 0.003 | | 0.035 | |
| *n* | 80 | | 80 | | 80 | | 80 | |

* Abbreviations: FITC-D, fluorescein isothiocyanate-dextran; SEM, Standard Error of the Means; L:M, Lactulose to Mannitol ratio; *n*, number of pens per effect. ** Several samples of lactulose lacked detectable limits in the serum. For analysis, these samples were assigned the arbitrary value of 1. [abc] Differences in superscripts within the column represent differences between groups.

**Table 2.** Effect of fed state on average serum sugar concentrations in broilers during a 42-day production cycle across 2 replicates.

| | FITC-D * | SEM | Mannitol | SEM | Lactulose ** | SEM | L:M | SEM |
|---|---|---|---|---|---|---|---|---|
| **Fed State** | **ng/mL** | | **µg/mL** | | **µg/mL** | | | |
| Fed | 86.21 [b] | 6.84 | 7.45 [a] | 1.34 | 2.10 [b] | 0.07 | 0.28 [b] | 0.08 |
| Fasted | 110.87 [a] | 8.31 | 5.57 [b] | 1.44 | 3.22 [a] | 0.11 | 0.58 [a] | 0.16 |
| *p*-value | 0.006 | | <0.001 | | <0.001 | | <0.001 | |
| *n* | 120 | | 120 | | 120 | | 120 | |

* Abbreviations: FITC-D, fluorescein isothiocyanate-dextran; SEM, Standard Error of the Means; L:M, Lactulose to Mannitol ratio; *n*, number of pens per effect. ** Several samples of lactulose lacked detectable limits in the serum. For analysis, these samples were assigned the arbitrary value of 1. [ab] Differences in superscripts within the column represent differences between groups.

**Table 3.** Effect of sugar supplemented on average serum sugar concentrations in broilers during a 42-day production cycle across 2 replicates.

| | FITC-D * | SEM | Mannitol | SEM | Lactulose ** | SEM | L:M | SEM |
|---|---|---|---|---|---|---|---|---|
| **Sugar Provided** | **ng/mL** | | **µg/mL** | | **µg/mL** | | | |
| No Sugar | 75.87 [b] | 8.16 | 3.81 [b] | 0.92 | 1.10 [b] | 0.05 | 0.29 [b] | 0.08 |
| FITC-D | 141.39 [a] | 13.71 | - | - | - | - | - | - |
| LMS | 89.01 [b] | 8.80 | 8.36 [a] | 2.03 | 4.17 [a] | 0.17 | 0.50 [a] | 0.14 |
| FITC-LMS | 95.69 [b] | 9.61 | 8.39 [a] | 2.03 | 3.85 [a] | 0.16 | 0.46 [a] | 0.12 |
| *p*-value | <0.001 | | <0.001 | | <0.001 | | <0.001 | |
| *n* | 60 | | 60 | | 60 | | 60 | |

* Abbreviations: FITC-D, fluorescein isothiocyanate-dextran; SEM, Standard Error of the Means; L:M, Lactulose to Mannitol ratio; LMS, Lactulose, Mannitol, Sucralose sugar treatment; FITIC-LMS, FITC-D and LMS combined sugar treatment; *n*, number of pens per effect. ** Several samples of lactulose lacked detectable limits in the serum. For analysis, these samples were assigned the arbitrary value of 1. - Values not determined. [ab] Differences in superscripts within the column represent differences between groups.

For the two-way interaction of fed state and sugar treatment, fasted-FITC-D, fasted-FITC-D+LMS, fasted-LMS, and fed-FITC-D had similar serum concentrations of FITC-D compared to each other ($p > 0.05$) but had greater serum FITC-D concentrations compared to fasted-Control, fed-FITC-D+LMS, fed-LMS, and fed-Control ($p < 0.05$; Table 4). The remaining two-way interactions and the three-way interaction were similar for serum FITC-D concentrations ($p > 0.201$; Tables 5 and 6; Figure 2A).

**Table 4.** Effect of the interaction between fed state and sugar supplemented on average serum sugar concentrations in broilers during a 42-day production cycle across 2 replicates.

| Fed State | Sugar Provided | FITC-D * | SEM | Mannitol | SEM | Lactulose ** | SEM | L:M | SEM |
|---|---|---|---|---|---|---|---|---|---|
| | | ng/mL | | µg/mL | | µg/mL | | | |
| Fed | No Sugar | 75.20 [b] | 11.20 | 4.64 | 1.15 | 1.00 [e] | 0.06 | 0.22 [d] | 0.06 |
| | FITC-D | 140.67 [a] | 19.80 | - | - | - | - | - | - |
| | LMS | 67.55 [b] | 9.51 | 9.07 | 2.25 | 2.86 [d] | 0.17 | 0.32 [c] | 0.09 |
| | FITC-LMS | 77.30 [b] | 10.97 | 9.82 | 2.44 | 3.25 [c] | 0.19 | 0.33 [c] | 0.09 |
| Fasted | No Sugar | 76.54 [b] | 11.23 | 3.14 | 0.78 | 1.21 [d] | 0.07 | 0.39 [c] | 0.11 |
| | FITC-D | 142.11 [a] | 17.38 | - | - | - | - | - | - |
| | LMS | 117.27 [a] | 15.04 | 7.70 | 1.91 | 6.06 [a] | 0.35 | 0.79 [a] | 0.22 |
| | FITC-LMS | 118.45 [a] | 15.68 | 7.17 | 1.78 | 4.56 [b] | 0.26 | 0.64 [b] | 0.18 |
| | *p*-value | 0.047 | | 0.193 | | <0.001 | | 0.046 | |
| | *n* | 30 | | 30 | | 30 | | 30 | |

\* Abbreviations: FITC-D, fluorescein isothiocyanate-dextran; SEM, Standard Error of the Means; L:M, Lactulose to Mannitol ratio; LMS, Lactulose, Mannitol, Sucralose sugar treatment; FITIC-LMS, FITC-D and LMS combined sugar treatment; *n*, number of pens per effect. \*\* Several samples of lactulose lacked detectable limits in the serum. For analysis, these samples were assigned the arbitrary value of 1. - Values not determined. [abcde] Differences in superscripts within the column represent differences between groups.

**Table 5.** Effect of the interaction between fed state and day of age on average serum sugar concentrations in broilers during a 42-day production cycle across 2 replicates.

| Fed State | Day of Age | FITC-D * | SEM | Mannitol | SEM | Lactulose ** | SEM | L:M | SEM |
|---|---|---|---|---|---|---|---|---|---|
| | | ng/mL | | µg/mL | | µg/mL | | | |
| Fed | 14 | 100.74 | 11.41 | 9.57 [a] | 2.37 | 2.55 [c] | 0.17 | 0.27 [d] | 0.78 |
| | 28 | 71.61 | 10.45 | 6.14 [bc] | 1.52 | 2.10 [d] | 0.15 | 0.34 [c] | 0.10 |
| | 42 | 88.82 | 10.08 | 7.04 [b] | 1.75 | 1.74 [e] | 0.10 | 0.25 [d] | 0.07 |
| Fasted | 14 | 136.69 | 14.85 | 5.89 [c] | 1.46 | 2.97 [bc] | 0.17 | 0.51 [b] | 0.14 |
| | 28 | 91.28 | 11.63 | 6.34 [bc] | 1.57 | 3.64 [a] | 0.21 | 0.57 [ab] | 0.16 |
| | 42 | 109.23 | 12.20 | 4.63 [d] | 1.15 | 3.09 [b] | 0.10 | 0.67 [a] | 0.07 |
| | *p*-value | 0.878 | | <0.001 | | <0.001 | | 0.004 | |
| | *n* | 40 | | 40 | | 40 | | 40 | |

\* Abbreviations: FITC-D, fluorescein isothiocyanate-dextran; SEM, Standard Error of the Means; L:M, Lactulose to Mannitol ratio; *n*, number of pens per effect. \*\* Several samples of lactulose lacked detectable limits in the serum. For analysis, these samples were assigned the arbitrary value of 1. [abcde] Differences in superscripts within the column represent differences between groups.

**Table 6.** Effect of the interaction between day of age and sugar supplemented on average serum sugar concentrations in broilers during a 42-day production cycle across 2 replicates.

| Day of Age | Sugar Provided | FITC-D * | SEM | Mannitol | SEM | Lactulose ** | SEM | L:M | SEM |
|---|---|---|---|---|---|---|---|---|---|
| | | ng/mL | | µg/mL | | µg/mL | | | |
| 14 | No Sugar | 88.87 | 13.89 | 3.32 [d] | 0.84 | 1.00 [d] | 0.07 | 0.30 [d] | 0.09 |
| | FITC-D | 178.54 | 26.87 | - | - | - | - | - | - |
| | LMS | 118.32 | 18.31 | 10.95 [a] | 2.77 | 4.34 [a] | 0.31 | 0.40 [dc] | 0.11 |
| | FITC-LMS | 101.00 | 15.36 | 11.63 [a] | 2.95 | 4.81 [a] | 0.35 | 0.41 [dc] | 0.12 |
| 28 | No Sugar | 59.41 | 12.57 | 2.82 [d] | 0.72 | 1.33 [c] | 0.10 | 0.47 [bc] | 0.13 |
| | FITC-D | 117.24 | 20.96 | - | - | - | - | - | - |
| | LMS | 84.15 | 15.00 | 8.80 [ab] | 2.23 | 4.67 [a] | 0.33 | 0.53 [bc] | 0.15 |
| | FITC-LMS | 72.87 | 14.70 | 9.78 [b] | 2.48 | 3.39 [b] | 0.25 | 0.35 [de] | 0.10 |
| 42 | No Sugar | 82.69 | 14.13 | 5.93 [c] | 1.52 | 1.00 [d] | 0.07 | 0.17 [f] | 0.05 |
| | FITC-D | 135.03 | 20.29 | - | - | - | - | - | - |
| | LMS | 70.82 | 11.17 | 6.05 [c] | 1.54 | 3.57 [b] | 0.27 | 0.59 [ab] | 0.17 |
| | FITC-LMS | 19.04 | 17.00 | 5.19 [c] | 1.30 | 3.49 [b] | 0.24 | 0.67 [a] | 0.19 |
| *p*-value | | 0.201 | | <0.001 | | <0.001 | | <0.001 | |
| *n* | | 20 | | 20 | | 20 | | 20 | |

\* Abbreviations: FITC-D, fluorescein isothiocyanate-dextran; SEM, Standard Error of the Means; L:M, Lactulose to Mannitol ratio; LMS, Lactulose, Mannitol, Sucralose sugar treatment; FITIC-LMS, FITC-D and LMS combined sugar treatment; *n*, number of pens per effect. \*\* Several samples of lactulose lacked detectable limits in the serum. For analysis, these samples were assigned the arbitrary value of 1. - Values not determined. [abcdef] Differences in superscripts within the column represent differences.

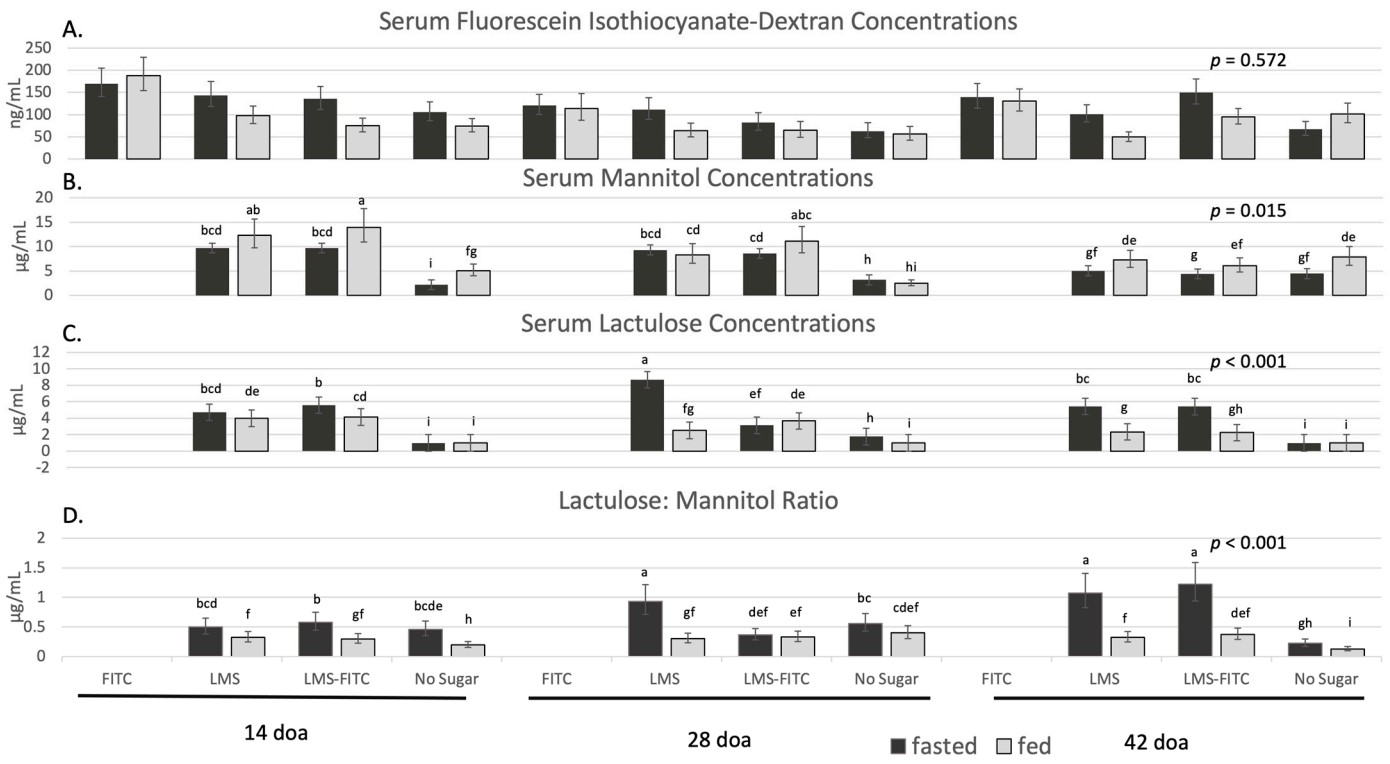

**Figure 2.** Average serum fluorescein isothiocyanate-dextran, mannitol, lactulose, and lactulose to mannitol ratio by fed state, age of bird, and sugar provided to the bird. Each bar represents the average serum concentrations from 10 pens. (**A**) Average concentration of serum fluorescein Isothiocyanate-dextran. (**B**) Average concentration of serum mannitol. (**C**) Average concentration of serum lactulose. (**D**) Average concentration of serum lactulose to mannitol ratio. Black bars represent the fasted condition with serum collected after a 12 h fast. Light grey bars represent the fed condition with serum collected with access to feed. Abbreviations: FITC-D, fluorescein isothiocyanate-dextran; LMS, Lactulose, Mannitol, Sucralose sugar treatment; FITIC-LMS, FITC-D and LMS combined sugar treatment. Differences in superscripts within the column represent differences between days of age, sugar provided to the bird, and fed state.

### 3.3. Serum Mannitol Concentrations

The effects of bird age, fed state, sugar treatment, the two-way interactions of bird age and fed state, bird age and sugar treatment, and the three-way interaction of bird age, fed state, and sugar treatment were all significant for serum mannitol concentrations ($p < 0.020$). Serum mannitol concentrations were elevated at 14 doa compared to 28 and 42 doa ($p < 0.050$; Table 1). Fed birds had elevated serum concentration of mannitol compared to fasted birds ($p < 0.010$; Table 2). Serum from birds given LMS had elevated serum mannitol compared to control birds that did not receive mannitol ($p < 0.010$; Table 3).

The two-way interaction of fed state and sugar treatment for serum mannitol concentrations was not significant ($p = 0.193$). With the two-way interaction of bird age and fed state, the highest serum mannitol concentrations were in fed birds at 14 doa; and the lowest serum mannitol concentration was observed in fasted birds at 42 doa ($p < 0.050$; Table 5). All other serum mannitol concentrations were found to be intermediate. With the two-way interaction of bird age and sugar treatment, the highest serum mannitol concentrations were in birds at 14 doa provided LMS and LMS-FITC, and broilers at 14 and 28 doa that did not receive any sugar treatment had the least serum mannitol concentrations ($p < 0.050$; Table 6). Differences in the three-way interaction are displayed in Figure 2B.

*3.4. Serum Lactulose Concentrations*

The effects of bird age, fed state, sugar treatment, and all two-way interactions and the three-way interaction of bird age, fed state, and sugar treatment were all significant for serum lactulose concentrations ($p < 0.003$). Serum concentrations of lactulose were elevated at 14 and 28 doa compared to 42 doa ($p < 0.050$; Table 1). Fasted birds had elevated serum concentration of lactulose compared to fed birds ($p < 0.05$; Table 2). Serum from birds given LMS and LMS-FITC had elevated serum lactulose compared to control birds that did not receive lactulose ($p < 0.050$; Table 3).

For the two-way interaction of sugar treatment and fed state, the lowest serum lactulose concentrations were observed in birds that did not receive any lactulose sugar from either the fed or fasted treatment. Fed birds provided lactulose had intermediate serum lactulose concentrations compared to fasted birds gavaged with lactulose, which had the highest serum lactulose concentrations ($p < 0.050$; Table 4). With the two-way interaction of bird age and fed state, the lowest serum lactulose concentrations were observed in fed birds at 42 doa, where fasted birds at 28 doa had the highest serum lactulose concentrations ($p < 0.050$; Table 5). With the two-way interaction of bird age and sugar treatment, the highest serum lactulose concentrations were observed in broilers at 14 doa gavaged with LMS and LMS-FITC or 28 doa gavaged with LMS. The lowest serum lactulose concentrations were found in broilers at 14, 28, and 42 doa who did not receive the lactulose sugar with many of these samples yielding below detectable levels ($p < 0.050$; Table 6). Differences in the three-way interaction are displayed in Figure 2C.

*3.5. Serum Lactulose to Mannitol Ratio*

The effect of bird age, fed state, sugar treatment, and all two-way interactions and the three-way interaction of bird age, fed state, and sugar treatment were all significant for the serum lactulose to mannitol ratio ($p < 0.047$). Serum concentrations of the lactulose to mannitol ratio were elevated at 28 and 42 doa compared to 14 doa ($p < 0.050$; Table 1). Fasted birds had an elevated serum lactulose to mannitol ratio compared to fed birds ($p < 0.05$; Table 2). Serum from birds given LMS and LMS-FITC had an elevated serum lactulose to mannitol ratio compared to control birds that did not receive mannitol and lactulose ($p < 0.050$; Table 3).

For the two-way interaction of sugar treatment and fed state, the highest serum lactulose to mannitol ratio was observed in fasted birds that received both lactulose and mannitol either as part of the LMS or LMS-FITC sugar treatment compared to all other groups ($p < 0.050$; Table 4). With the two-way interaction of bird age and fed state, the fed state played a significant role with all days of the fed group having a lower serum lactulose to mannitol ratio than all of the fasted group. The lowest serum lactulose to mannitol ratio was observed in fed birds at 42 and 14 doa, whereas fasted birds at 42 doa had the highest serum lactulose to mannitol ratio ($p < 0.050$; Table 5). Concerning the two-way interaction of bird age and sugar, while a significant interaction was found, no differences between sugar treatments were observed at 14 doa ($p > 0.050$). However, at 28 doa, birds that received no sugar or received LMS had higher serum lactulose to mannitol ratio concentrations compared to LMS-FITC. At 42 doa, serum concentrations of the lactulose to mannitol ratio were similar for LMS-FITC and LMS and were greater than in birds that received no sugar ($p < 0.050$; Table 6). Differences in the three-way interaction are displayed in Figure 2D.

## 4. Discussion

Intestinal health is a concern in the poultry industry as it encompasses anything from enteric disease to reduced performance. Because poultry producers aim to maximize production and efficiency, new methods for monitoring and measuring sub-clinical departures in intestinal health are crucial. Unlike other species, the lack of urine excretion has limited non-lethal methods of determining intestinal health, particularly intestinal permeability from directly translating across other species. This study aimed to examine the effectiveness

of commonly used methods to determine intestinal permeability in livestock across the broiler production cycle.

First, the use of FITC-D to measure intestinal barrier permeability has been widely used in poultry research and can detect extreme changes in intestinal permeability. This relatively simple to perform assay requires the use of a fluorescently labeled maker that can immediately be read using a fluorescent plate reader and has great potential for use in a commercial facility due to ease of administering and measuring. The present study identified that birds which were gavaged with FITC-D exhibited the highest relative serum fluorescence compared to birds that did not receive FITC-D. However, it should be noted that control birds had relatively high fluorescence levels demonstrating a high level of autofluorescence of the serum. This has been observed in other studies and was an underlying need for optimization [17,18,21]. As expected, broilers that were fasted also had higher serum fluorescence compared to fed broilers. The ability to detect these changes was supported by Baxter et al. [17] who used fasting as a method to optimize the FITC-D protocol. When comparing the two-way interaction of fed state and sugar provided, we observed that birds provided FITC-D had higher fluorescence regardless of fed state, indicating that the assay could not detect a state of expected intestinal permeability. Additionally, both LMS (containing no FITC-D) and FITC-LMS gavaged birds had elevated fluorescence in the fasted birds compared to fed birds. While these results disagree with other reports [3,4,14,17,18], where serum FITC-D concentrations were elevated in fasted broilers provided FITC-D, it should be noted that these studies did not have the two-way interaction of fed state and sugar treatment. These studies only compared fed versus fasted with all birds receiving FITC-D [3,4,17]. This is comparable to the main effect of dietary treatment in this study where a difference was observed statistically. While a numerical increase in FITC-D in the serum was observed with fasted birds compared to control fasted birds, the inability of this study to detect these differences is likely a statistical issue. The standard error of the means in this study ranged between 6.84 and 20.96 ng/mL despite a minimum of 40 birds used per treatment within the three-way interaction. Therefore, the inability of FITC-D to detect differences in fed state and sugar treatment for the FITC-D birds and the high variation makes it a less appealing method to be used in commercial settings as small differences could not be detected. Additionally, this method would not provide consistent differences unless a flock is undergoing a major enteric challenge.

Second, mannitol is a small monosaccharide used in combination with the disaccharide, lactulose, to determine intestinal permeability in urine of many mammalian species. In using the lactulose to mannitol ratio, it is expected that mannitol will pass at a constant rate across the intestinal epithelium of all birds administered mannitol. However, this was not observed in this study. In this study, a greater concentration of mannitol in fed birds was observed compared to fasted. Additionally, mannitol concentrations were observed in broilers that did not receive mannitol. Comparison of differences in collection of urine versus blood highlight the fact that urine is a filtered product where many substrates, particularly glucose, are reabsorbed into the blood [22]. Therefore, glucose and other small molecules would be included in blood but not urine. Furthermore, comparisons of the molecular size of glucose (180 mol/g) to mannitol (183 mol/g) and the chemical composition of glucose ($C_6H_{12}O_6$) to mannitol ($C_6H_{14}O_6$) demonstrate that the differences are relatively minimal. Additionally, an ion chromatograph run spiked with glucose showed glucose elutes near the same time as mannitol, confirming this suspicion (analysis not shown). This, along with the poor chromatogram peak separation of mannitol, might indicate that the elevated "mannitol" concentrations likely include glucose. Based on these data, the use of mannitol and the subsequent lactulose to mannitol ratio would not be advised in birds unless methods are used that improve the elution of mannitol from other substances.

Third, lactulose is a disaccharide molecule, is larger than mannitol, and is expected to pass through the intestinal barrier when intestinal stress is elevated, unlike mannitol. Lactulose is expected to pass through the intestinal barrier at a similar rate to FITC-D.

Therefore, it is not surprising that this method had similar results to the results observed with FITC-D administration, with both being elevated in serum during fasting. Unlike with the serum concentration of FITC-D, this study observed the majority of samples from control broilers having no detectible concentrations of lactulose in serum, demonstrating the selectivity of the assay. This is essential when determining if the assay can be used for determination of intestinal permeability in commercial broiler production. The lack of detection of lactulose in birds not administered lactulose establishes a clean background unlike the FITC-D assay which has a high background signal. Additionally, the ability to observe an average 73.8% increase in lactulose in fasted and fed birds across the sugar treatments is a greater percent increase compared to the FITC-D assay (19.5%). Unlike the FITC-D assay, the lactulose assay produced consistent differences between fasted and fed birds across different ages, indicating the potential robustness of this assay for a commercial setting. However, this study consisted of two replications in non-commercial settings. Additional studies would need to be conducted to determine if other known intestinal stressors can elicit a similar result and if a range can be established to determine optimal, sub-optimal, and detrimental concentrations of lactulose. Lastly, it should be mentioned that when comparing differences in the methods, determination of serum lactulose does require significantly more time, expertise, and equipment to determine serum concentrations due to the use of the ion chromatograph.

While age was not a major focus of this experiment, age remains a consideration when trying to understand intestinal permeability and determine optimal methods to measure intestinal permeability. In this study, we observed a significant and consistent decrease in all markers of intestinal permeability at 28 doa in both trials. The concentration of sugars observed at 14 doa indicated a higher degree of intestinal permeability in younger birds. This may be due to the fact that the gastrointestinal tract and intestinal microbiota are not fully matured and developed, respectively, during the earlier weeks of life. Interestingly, serum concentrations of lactulose increased at 42 doa compared to 28 doa. These trends were observed over both trials leading to questions about what might be occurring in the broiler between 28 and 42 doa that leads to these differences. These results were not expected but may be explained by shifts in the intestinal microbiota. A study by Lu et al. [23] identified changes in intestinal microbiota at various ages in broilers. They discovered that microbial populations and microbial community structure were the most stable during 14 to 28 days of age and then had significant shifts around 49 days of age. An additional study by Liao et al. [24] confirms changes in intestinal morphology in growing broilers. Increased permeability at day 42 may also result from general stress or heat stress caused by heavy body weights during the finisher period [25,26].

## 5. Conclusions

In conclusion, this study examined serum concentrations of FITC-D, mannitol, lactulose, and the lactulose to mannitol ratio during the 6-week production cycle in two different flocks of broilers to determine intestinal permeability as an indicator of intestinal health. As anticipated, the study observed significant increases in FITC-D and lactulose with fasted compared to fed states and when the sugar was provided compared to the control. This unfortunately was not the observation with mannitol. In fact, the opposite was observed. Given this assay was conducted with serum, it is suspected that the high concentrations of circulating glucose are co-eluting with mannitol and altering the value associated with mannitol concentrations with detection in blood for intestinal permeability studies in poultry. This would suggest that the mannitol and the subsequent lactulose to mannitol ratio are invalid assays in poultry using the column separation methodology used in this study. Lactulose assays reveal lower serum concentration of lactulose during fed states and non-detectable concentrations when not given. While the use of lactulose demonstrates a promise for the use in a commercial setting, additional studies are needed to determine the effects of other stressors on serum lactulose and if ranges can be established to determine optimal versus sub-optimal conditions in commercial broiler facilities.

**Supplementary Materials:** The following supporting information can be downloaded at: https://www.mdpi.com/article/10.3390/poultry2030028/s1, Table S1: Diet Composition for both broiler replications.

**Author Contributions:** M.L.W. contributed to data collection, analysis, writing, and reviewing of the manuscript. B.J.K. and D.A.K. contributed research funds, experimental design, data collection, analysis, writing, and review of the manuscript. All authors have read and agreed to the published version of the manuscript.

**Funding:** This work was supported by the Iowa Agriculture and Home Economics Experiment Station (IAHEES) Project Proposal for Project No. IOW04100 and by appropriated funds from USDA-ARS CRIS 5030-31000-007-00D.

**Institutional Review Board Statement:** All procedures involving live animals were approved by the Iowa State University Institutional Animal Care and Use Committee (IACUC number 18-331).

**Informed Consent Statement:** Not applicable.

**Data Availability Statement:** Data are available upon request.

**Acknowledgments:** The authors would like to thank Jennifer Cook for her contributions in troubleshooting and sample analysis with the ion chromatograph.

**Conflicts of Interest:** The authors declare no conflict of interest.

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
