# Peer review of "Comparison of Intestinal Permeability Methods in Broilers over a 6-Week Growth Period"

_poultry, doi:10.3390/poultry2030028_

Round 1

Reviewer 1 Report

The study of intestinal permeability measurement method has important academic value. The experimental design of this paper is reasonable. But there are a few problems:

1. The introduction of the method in the abstract part is too detailed, and the conclusion part should be compressed. you should simplify them.

2. Please explain in the discussion: why FITC-D was detected in the control without intaking FITC-D? Whether it was equipment contamination or test operation error?

3. It is suggested to change the title to "comparative study of intestinal permeability evaluation methods in broilers over a 6-week growth period"

4. Line 18: "Serum FITC-D had a significant sugar by fed state interaction"?

5. Line 380: should be "in blood"?

1. poor legibility, and there are some astringent sentences;

2. some descriptions are not appropriate. for example, "compared with 42 days of age, xxxx increased at 28 days of age". It is better to write "compared with 28 days of age, xxxx decreased at 42 days of age".

Author Response

Thank you very much for you time and expertise in reviewing of this manuscript.

  1. The introduction of the method in the abstract part is too detailed, and the conclusion part should be compressed. you should simplify them. As the methods are the main focus of the manuscript, the amount of detail is needed to provide a potential reader with necessary details to determine if they wish to read the full paper or find other papers to read. The conclusion of the abstract is a single sentence and can not be compressed.
  2. Please explain in the discussion: why FITC-D was detected in the control without intaking FITC-D? Whether it was equipment contamination or test operation error? The explanation can be found in Lines 350-352. Briefly, serum autofluorescence occurs at similar wavelength as FITC-D and therefore provided high background noise.
  3. It is suggested to change the title to "comparative study of intestinal permeability evaluation methods in broilers over a 6-week growth period" Changed
  4. Line 18: "Serum FITC-D had a significant sugar by fed state interaction"? It has been rephrased to say “Serum FITC-D only had a significant sugar by fed state interaction (P>0.05), whereas serum lactulose was significant for all interactions.
  5. Line 380: should be "in blood"? It is unclear where in the sentence the phrase “in blood” should be. Line 380 reads “in broilers that did not receive mannitol. Comparison of differences in collection of urine”.

poor legibility, and there are some astringent sentences.  We are seeking clarity from these statements.  Our understanding of “legibility” based on the Cambridge Dictionary is,” the degree to which writing or text can be read easily because the letters are clear, the text is printed well, etc.:” Given our document was typewritten in the recommended font type (Palatino Linotype), we are uncertain as to how our text is of “poor legibility”.  Additionally, our understanding of the word “astringent” again based on Cambridge Dictionary is, “a substance that causes the skin or other tissue to tighten, or remarks that are clever but unkind or criticize someone”.   We have re-examined the manuscript and did not find any unkind or criticizing words toward a particular group. The intent of the manuscript is to provide factual and not unkind or criticizing words. We do not wish to be unkind or criticize any of the researchers, we are requesting the reviewer provide clarity and specific examples.

  1. some descriptions are not appropriate. for example, "compared with 42 days of age, xxxx increased at 28 days of age". It is better to write "compared with 28 days of age, xxxx decreased at 42 days of age". Thank you for the suggestion.

Reviewer 2 Report

In this study, two broiler stress models (fasting and sugar treatment) were established to compare the differences in intestinal permeability biomarkers (FITC-D and LMS), and it was found that FITC-D, lactulose and mannitol differed under different stress models. A new idea for the commercial determination of intestinal permeability in broiler chickens is provided. The manuscript benefits from a detailed background of the research on intestinal permeability and a rational design of the broiler stress test. This manuscript is innovative, but needs some additional revisions.

Question 1.

Line 2: The title needs to be revised, it does not fit well with the test content

Question 2.

Line 8-22: Summary lacks a conclusion

Question 3

line77: 12 individuals with low numbers, lack of diet composition and nutritional level

Question 4.

Line 99: Figure1 does not seem very clear

Question 5.

The data in the manuscript are reliable, but the background of the test chickens is small. The manuscript could be more convincing if the test chickens' weight and other production performance indicators were added, and if subsequent biomarker verification experiments in broiler serum were added.

Author Response

Thank you very much for your time and expertise in reviewing the manuscript improve it.

Question 1.

Line 2: The title needs to be revised, it does not fit well with the test content Changed.

Question 2.

Line 8-22: Summary lacks a conclusion Added.

Question 3

line77: 12 individuals with low numbers, lack of diet composition and nutritional level. The number of individual birds per pen is set by industry and the Ag Guidelines.  As stated in linen 79, our experimental unit is the pen, and we repeated the experiment over 2 replications. Experimental unts range from 10 to 40 depending on the variable tested. We have added the diet composition as a supplemental table.

Question 4.

Line 99: Figure1 does not seem very clear. We have corrected the figure legend in the figure to correctly identify the LMS treatment, and added the following description to the figure description, “Sugar treatments include Control which were provided water; FITC-D which were provided fluorescein isothiocyanate-dextran at a rate of 8.32 mg FITC-D/kg BW; LMS which were provided lactulose, mannitol, and sucralose at a rate of 0.25 g lactulose/kg BW, 0.05 g mannitol/kg BW, and 0.05 g sucralose/kg BW; and FITC-LMS with was a combination of FITC-D and LMS at rates described above.”

Question 5.

The data in the manuscript are reliable, but the background of the test chickens is small. The manuscript could be more convincing if the test chickens' weight and other production performance indicators were added, and if subsequent biomarker verification experiments in broiler serum were added. Animal parameters provide include average body weights and mortality (see lines 170-178). Biomarker verification assays were conducted by previous researchers and thus we used optimized methods and did not conduct formal verification experiments. We did conduct internal optimization for equipment of the protocols, however; we do not feel they are appropriate for publication.

Reviewer 3 Report

The objective of this study was to examine the intestinal permeability of broilers. The research involved conducting experiments to compare the efficacy of serum FITC-D and serum LMS in assessing intestinal permeability in poultry under varying feeding conditions. Moreover, the researchers observed notable increases in FITC-D and lactulose levels during fasting, while no significant changes were observed in mannitol levels between fed and fasted states. These findings provide valuable insights into the use of lactulose assays as a means to determine intestinal permeability, thereby serving as an indicator of intestinal health.

Author Response

Thank you very much for your time and expertise in reviewing the manuscript. 

Round 2

Reviewer 1 Report

1. The introduction of the method in the abstract part is too detailed. suggest deleting the method of detection.  

2. Please explain in the discussion: why FITC-D was detected in the control without intaking FITC-D? Whether it was equipment contamination or test operation error? I did not find the explanation in Lines 350-352, which says "First, the use of FITC-D to measure intestinal barrier permeability has been widely used 350 in poultry research and can detect extreme changes in intestinal permeability. This rela- 351 tively simple assay to perform requires the use of a fluorescently labeled maker that can 352 immediately be read using a fluorescent plate reader and has great potential for use in a 353 commercial facility due to ease of administering and measuring. "

3. "Serum FITC-D only had a significant sugar by fed state interaction (P<0.05), whereas serum lactulose 19 was significant for all interactions with elevated concentrations in broilers provided lactulose and 20 fasted (P<0.001)". I could not understand the meaning of this sentence, especially "a significant sugar".

 I think the manuscript is difficult to read for non-native speakers.

Author Response

  1. The introduction of the method in the abstract part is too detailed. suggest deleting the method of detection.  We have removed the text.
  2. Please explain in the discussion: why FITC-D was detected in the control without intaking FITC-D? Whether it was equipment contamination or test operation error? I did not find the explanation in Lines 350-352, which says "First, the use of FITC-D to measure intestinal barrier permeability has been widely used 350 in poultry research and can detect extreme changes in intestinal permeability. This rela- 351 tively simple assay to perform requires the use of a fluorescently labeled maker that can 352 immediately be read using a fluorescent plate reader and has great potential for use in a 353 commercial facility due to ease of administering and measuring. " In the updated version, the lines are 348-353 and contain the following text. “The present study identified that birds which were gavaged with FITC-D exhibited the highest relative serum fluorescence compared to birds that did not receive FITC-D. However, it should be noted that control birds had relatively high fluorescence levels demonstrating a high level of autofluorescence of the serum. This has been observed in other studies and was an underlying need for optimization [17,18,21].” We did find that one of the versions uploaded contained track changes and shifted the lines and caused the discrepancy.
  3. "Serum FITC-D only had a significant sugar by fed state interaction (P<0.05), whereas serum lactulose 19 was significant for all interactions with elevated concentrations in broilers provided lactulose and 20 fasted (P<0.001)". I could not understand the meaning of this sentence, especially "a significant sugar". Given the additional space, we have changed the text to be, “Serum FITC-D only had a significant sugar by fed state interaction (P<0.05) with elevated concentrations in fasted and fed birds that received FITC-D. Serum lactulose was significant for all interactions with elevated concentrations in broilers provided lactulose and fasted (P<0.001).”

Comments on the Quality of English Language  I think the manuscript is difficult to read for non-native speakers. We agree that this is a difficult manuscript due to the complexity of the experimental design.  We greatly appreciate your willingness to go through this manuscript with scientific and English rigor.